The dynamics of circulating SARS-CoV-2 lineages in Bogor and surrounding areas reflect variant shifting during the first and second waves of COVID-19 in Indonesia

Prasetyoputri Anggia 1
Dharmayanthi Anik B. 2
Iryanto Syam B. 3
Andriani Ade 1
Nuryana Isa 1
Wardiana Andri 1
Ridwanuloh Asep M. 1
Swasthikawati Sri 1
Hariyatun Hariyatun 1
Nugroho Herjuno A. 2
Idris Idris 2
Indriawati Indriawati 1
Noviana Zahra 2
Oktavia Listiana 4
Yuliawati Yuliawati 1
Masrukhin Masrukhin 2
Hasrianda Erwin F. 2
Sukmarini Linda 1
Fahrurrozi Fahrurrozi 1
Yanthi Nova Dilla 1
Fathurahman Alfi T. 1
Wulandari Ari S. 1
Setiawan Ruby 2
Rizal Syaiful 2
Fathoni Ahmad 1
Kusharyoto Wien 1
Lisdiyanti Puspita 1
Ningrum Ratih A. 1
Saputra Sugiyono sugiyono.saputra@brin.go.id 2
1 Research Center for Biotechnology, National Research and Innovation Agency (BRIN) , Bogor , West Java , Indonesia
2 Research Center for Biology, National Research and Innovation Agency (BRIN) , Bogor , West Java , Indonesia
3 Research Center for Informatics, National Research and Innovation Agency (BRIN) , Bogor , West Java , Indonesia
4 Research Center for Chemistry, National Research and Innovation Agency (BRIN) , Bogor , West Java , Indonesia
Mossong Joël
Electronic publication date: 2022 Mar 22
Publication date: 2022
Volume: 10
Electronic Location ID: e13132
Received 2021 Nov 11; Accepted 2022 Feb 26
Copyright: ©2022 Prasetyoputri et al.
Copyright year: 2022
Copyright holder: Prasetyoputri et al.
License: This is an open access article distributed under the terms of the Creative Commons Attribution License, which permits unrestricted use, distribution, reproduction and adaptation in any medium and for any purpose provided that it is properly attributed. For attribution, the original author(s), title, publication source (PeerJ) and either DOI or URL of the article must be cited.
License URL: https://creativecommons.org/licenses/by/4.0/

Keywords: SARS-CoV-2, COVID-19, Indonesian lineages, Bogor, Variant shifting, Nanopore, Genomic surveillance, Delta variants

Funding: DIPA IPH LIPI 2020 Research Grant B-5114/IPH/HK.01.03/I/2020 RISPRO-LPDP Funding Program for COVID-19 07/FI/P-KCOVID-19.2B3/X/2020 “Surveilans Genom SARS-CoV-2” (VenomCoV) LIPI/BRIN This work was supported by the DIPA IPH LIPI 2020 Research Grant (B-5114/IPH/HK.01.03/I/2020) and the RISPRO-LPDP Funding Program for COVID-19 (07/FI/P-KCOVID-19.2B3/X/2020) under the project name “Surveilans Genom SARS-CoV-2” (VenomCoV) LIPI/BRIN. The funders had no role in study design, data collection and analysis, decision to publish, or preparation of the manuscript.

==============================
Background

Indonesia is one of the Southeast Asian countries with high case numbers of COVID-19 with up to 4.2 million confirmed cases by 29 October 2021. Understanding the genome of SARS-CoV-2 is crucial for delivering public health intervention as certain variants may have different attributes that can potentially affect their transmissibility, as well as the performance of diagnostics, vaccines, and therapeutics.

Objectives

We aimed to investigate the dynamics of circulating SARS-CoV-2 variants over a 15-month period in Bogor and its surrounding areas in correlation with the first and second wave of COVID-19 in Indonesia.

Methods

Nasopharyngeal and oropharyngeal swab samples collected from suspected patients from Bogor, Jakarta and Tangerang were confirmed for SARS-CoV-2 infection with RT-PCR. RNA samples of those confirmed patients were subjected to whole genome sequencing using the ARTIC Network protocol and sequencer platform from Oxford Nanopore Technologies (ONT).

Results

We successfully identified 16 lineages and six clades out of 202 samples (male n = 116, female n = 86). Genome analysis revealed that Indonesian lineage B.1.466.2 dominated during the first wave (n = 48, 23.8%) while Delta variants (AY.23, AY.24, AY.39, AY.42, AY.43 dan AY.79) were dominant during the second wave (n = 53, 26.2%) following the highest number of confirmed cases in Indonesia. In the spike protein gene, S_D614G and S_P681R changes were dominant in both B.1.466.2 and Delta variants, while N439K was only observed in B.1.466.2 (n = 44) and B.1.470 (n = 1). Additionally, the S_T19R, S_E156G, S_F157del, S_R158del, S_L452R, S_T478K, S_D950N and S_V1264L changes were only detected in Delta variants, consistent with those changes being characteristic of Delta variants in general.

Conclusions

We demonstrated a shift in SARS-CoV-2 variants from the first wave of COVID-19 to Delta variants in the second wave, during which the number of confirmed cases surpassed those in the first wave of COVID-19 pandemic. Higher proportion of unique mutations detected in Delta variants compared to the first wave variants indicated potential mutational effects on viral transmissibility that correlated with a higher incidence of confirmed cases. Genomic surveillance of circulating variants, especially those with higher transmissibility, should be continuously conducted to rapidly inform decision making and support outbreak preparedness, prevention, and public health response.

Introduction

First detected in Wuhan, China in December 2019, a novel coronavirus named as the Severe Acute Respiratory Syndrome Coronavirus 2 (SARS-CoV-2) is the causative agent of Coronavirus Disease 2019 (COVID-19). COVID-19 cases rapidly increased to reach an epidemic scale, and was therefore announced as a Public Health Emergency of International Concern (PHEIC) on January 31st, 2020 and subsequently characterized as a pandemic on March 11th, 2020 (WHO, 2020). This status indicated that multiple countries may pose risks and an international response is required (Li et al., 2020b). According to the World Health Organization (WHO) (WHO, 2021b), a total of 246 million COVID-19 cases have been detected worldwide (as of 25 October 2021) with approximately 2.1% mortality rate. Since the first case was reported in March 2020, Indonesia had reached 1 million cases by late January 2021 with a mortality rate of 2.7% (BNPB, 2021). In July 2021, Indonesia experienced the second wave, which reached the highest peak since the beginning of the pandemic. More than 56,000 cases in one day were then recorded, with a cumulative number of 4.2 million cases by end of October 2021 (COVID-19 S, 2021). The two provinces with the highest cases were DKI Jakarta and West Java, which accounted for 20% and 16% of national cases, respectively (COVID-19 S, 2021).

Genetic characterization of SARS-CoV-2 is critical to enable monitoring of viral evolution and its epidemiology. Information on its circulating variants is useful to determine potential markers of transmissibility, altered antigenicity or severity of disease. The role of sequence data availability will become increasingly important as SARS-CoV-2 vaccines and antivirals become available, as it informs on circulating variants that can be inhibited by the vaccines and the possible emergence of antiviral resistance (Nasir et al., 2020; Takenouchi et al., 2021).

In October 2021, there are more than 4.7 million SARS-CoV-2 published sequences worldwide through the Global Initiative on Sharing All Influenza Data (GISAID) and those have been classified into nine large clades (GISAID, 2021). There were five major lineages (A, B, B.1, B.1.1, and B.1.177) that have been proposed by Phylogenetic Assignment of Named Global Outbreak LINeages (PANGOLIN) (Rambaut et al., 2020) but it has since been expanded into numerous lineages circulating globally. Furthermore, WHO has labeled several variants that pose an increased risk to global public health with Variants of Interest (VOIs) and Variants of Concern (VOCs). In this study, we aimed to identify circulating variants of SARS-CoV-2 in Bogor, West Java and surrounding areas as those cities also experienced massive waves of COVID-19 cases during the pandemic.

Materials & Methods

Ethical clearance/statement

The study protocol was reviewed and approved by the Health Research Ethics Committee, University of Indonesia and Cipto Mangunkusumo Hospital (HREC-FMUI/CMH) (20-10-1321_EXP). All oropharyngeal and nasopharyngeal swab samples used in this study are those accompanied by written informed consents signed by the patients, agreeing to donate their samples for research purposes.

Sample collection, RNA extraction and qRT-PCR

The study was carried out at Biosafety Level-3 and Biosafety Level-2 Laboratory, Cibinong Science Center, National Research and Innovation Agency (BRIN; formerly known as Indonesian Institute of Sciences or LIPI). Samples were oropharyngeal and nasopharyngeal swabs obtained from various clinics, hospitals, and diagnostic laboratories in Bogor (Bogor City and Bogor Regency), Jakarta and Tangerang (Tangerang and Tangerang Selatan). Bogor and Tangerang have a close proximity to Jakarta and a great proportion of its residents commute to work in Jakarta, the capital city of Indonesia. Nasopharyngeal and oropharyngeal swab samples were collected between May 2020 and August 2021, which were sent to BSL-3 and BSL-2 Lab for qRT-PCR confirmation of SARS-CoV-2 infection. Viral RNA was extracted using Viral Nucleic Acid Extraction (Geneaid Biotech Ltd., Taiwan) according to the manufacturer’s instructions. Amplification analysis was conducted using CE IVD real time PCR kit (Real Q 2019-nCoV Detection Kit Biosewoom, Korea) with envelope (E) and RNA dependent RNA polymerase (RdRp) as targeted genes and human RNAse P as internal control. The kit cut-offs were set at cycle threshold (Ct) ≤38 (E and RdRp) and ≤35 (RNase P) with limit of detection (LOD) sensitivity of 6.25 copies/µL for sputum and 3.125 copies/µL for swab in CFX 96 BioRad RT-PCR instrument. The qRT-PCR conditions were as follows: 50 °C for 30 min (1 cycle), 95 °C for 15 min (1 cycle), 95 °C for 15 s and 62 °C for 45 s (40 cycles).

Library preparation

SARS-CoV-2 viral RNA from swab specimens confirmed by qRT-PCR with Ct values of 11-30 were selected (File S1). Following a decrease of COVID-19 cases after August 2021, many confirmed positive samples were not qualified to be subjected for whole genome sequencing due to higher Ct values. Samples with Ct values <15 were diluted 10 times with nuclease-free water. Library preparation was performed according to the ARTIC nCoV-2019 sequencing protocol V2 (Quick, 2020) and then V3 after protocol update was released. Briefly, the complementary DNA (cDNA) was synthesized using SuperScript IV Reverse Transcriptase and either random hexamers (NEB) (V2) or LunaScript® RT SuperMix Kit (V3). cDNA was amplified by touch-down PCR (the annealing temperature is decreased by 0.1 °C after every cycle) using nCoV-2019/V3 primers (Itokawa et al., 2020) (IDT) to generate overlapping 400 base pair (bp) amplicons covering all the ∼30 Kbp SARS-CoV-2 genomes. A modification had been applied in V3 protocol by adding a primer pair that covers the #74 amplicon (0.3 µl, final concentration 2.5 pmol) in pool B reaction. Subsequently, crude PCR amplicons were cleaned up using AMPure XP beads (Beckman) in a 1:1 ratio. The purified amplicons were then quantified using the Qubit High Sensitivity double strand DNA (dsDNA) assay kit and measured on a Qubit 2.0 instrument (ThermoFisher Scientific). A total input DNA of 50 ng per sample was applied to a one-pot native barcoding reaction and subsequently was subjected to an end prep reaction using NEBNext Ultra II End Prep kit (NEB), followed by native barcoding using NB Expansion Kit 1-24 (Oxford Nanopore Technologies, ONT). Up to 23 samples (including a negative control) were barcoded and sequenced in any single run. All barcoded amplicons were then pooled and subjected to AMPure XP beads purification and Qubit quantification. Adapter ligation was then performed using AMII adapter mix (ONT) and T4 DNA ligase (NEB). Following another AMPure XP beads clean up and Qubit quantification, the final library was put on ice until use. Approximately 15 ng of final library were loaded onto an R9.4.1 flow cell mounted on a MinION Mk1b/Mk1c or PromethION sequencer (ONT).

Sequencing and bioinformatics analysis

Sequencing run was performed using MinKNOW (v20.06.4) with fast basecalling mode and the sequencing output was monitored in real time using RAMPART (v2.1.0) software package. The run was terminated when the coverage of sequences achieved >99% compared to the Wuhan-Hu-1 reference genome (MN908947.3). A complete bioinformatics ARTIC v1.1.0 (Loman et al., 2020) was used to generate consensus genome sequences of SARS-CoV-2 with default parameters. Guppy (v 4.0.14, ONT) with a high-accuracy mode was used to perform basecalling to obtain fastq files and classified the basecalled reads based on both barcodes. The reads were then aligned to the Wuhan-Hu-1 reference genome (https://www.ncbi.nlm.nih.gov/nuccore/MN908947.3) using minimap2 (v 2.10-r761) (Li, 2018) and trimmed to remove the primer sequences from the end-read alignments. The filtered reads were then used to call the variant and build a consensus SARS-CoV-2 sequence using Medaka (v 1.0.3) workflows (ONT) and bcftools (v 1.10.2) (Li, 2011), respectively.

Lineage, variant calling and phylogenetic analysis

Lineages of 202 SARS-CoV-2 genomes were identified using PANGOLIN v3.1.16 with designated pangoLEARN version v1.2.105 (O’Toole et al., 2020; O’Toole et al., 2021). The phylogenomic tree (Maximum Likelihood; GTR+F+I evolutionary model) was build using Augur utility (github.com/Nextstrain/augur) from the Nexstrain bioinformatic pipeline (Hadfield et al., 2018), employing MAFFT (Katoh & Standley, 2016) and IQ-TREE (Nguyen et al., 2015). The ultrafast bootstrap (UFBoot) (Minh, Nguyen & Von Haeseler, 2013) was performed and evaluated with 1,000 bootstrap replications. Unless mentioned otherwise, all plots were constructed using R v 4.1.2 (Team RC, 2021) as uploaded to GitHub (https://github.com/zahranoviana/VenomCov-Figures/tree/main).

Results

Patient demographics

Clinical samples (n = 202) were collected between May 2020 and August 2021 from various hospitals, diagnostic laboratories, and healthcare facilities in Bogor and its surrounding areas (Bogor n = 141, Jakarta n = 51 and Tangerang n = 10) as shown in Figs. 1 and 2A. Patients were male (57%) and female (43%) with the dominant age group being adults (35–64 years old; 52.6% and 58.1%), followed by young adults (20–34 years old; 33.6% and 30.2%). (Figs. 2A–2B).

Figure 1 A map of locations from which samples were obtained in Bogor and surrounding areas.

Samples were obtained from laboratories located in Bogor (1 and 2), Tangerang (3 and 4) and Jakarta (5), and subsequently subjected to qRT-PCR confirmation and whole genome sequencing at the Biosafety Laboratory Level 2 and 3 BRIN, Cibinong Science Center-Botanical Garden.

Figure 2 Summary of the patients’ profiles.

Composition of males and females are shown across sampling locations (A), age groups (B) and lineage distribution (C) from the 202 SARS-CoV-2 genomes sequenced. Others in male lineages (5.2%, 6/116) include B.56 (n = 2), B.1.36.19, AY.39, AY.42 and AY.79; while others in female lineages (5.8%, 5/86) include B.1.1, B.1.1.53, B.1.1.10, AY.43 and unclassified.The samples were mainly obtained from originating laboratory in Bogor (69.8%). Over 50% of samples in both males and females were from patients aged 35–64 years. A higher proportion of lineages B.1.1.398 and B.1.470 were observed in male patients compared to females, whereas lineages B.1 and B.1.459 were more prevalent in females compared to males.

Variations of lineages

The SARS-CoV-2 genome was amplified using nCoV-2019/V3 primers resulting in 400 bp amplicons that covers 28,999–29,782 bp. All viral genomes have been published in GISAID (File S1). Using PANGOLIN phylogenetic classification (Rambaut et al., 2020), the 201 sequenced genomes were determined to belong to 17 lineages. One virus (hCoV-19/Indonesia/JK-LIPI135/2021) was categorized as unclassified based on PANGOLIN but clustered together with Delta variants in clade GK by GISAID (stretches of NNNs 3.91% of overall sequences). We found that the majority of genomes of samples collected belonged to Delta variants or B.1.617.2 descendant (n = 49, 24.3%); Indonesian lineage B.1.466.2 (n = 48, 23.8%), B.1.1.398 (n = 29, 14.4%), B.1.470 (n = 22, 10.9%), B.1.459 (n = 18, 9.9%); European lineage B.1 (n = 22, 10.9%). In this study, the Delta variants include the lineage from AY.23 (n = 41, 20.3%), AY.24 (n = 4, 2%) as well as one virus of AY.39, AY.42, AY.43 and AY.79 lineage. The phylogenetic tree of all viruses is presented in Fig. 3.

Figure 3 Phylogenetic tree from 202 viruses, representing 17 lineages and six GISAID clades.

The clustering of certain lineages is indicative of the extent of genomic variations within the genomes. Three major clades were observed, including GH, GR and GK, while the minor clades were L, O and G. Clades GH and GR include several Indonesian lineages, while clade GK consists of Delta variants (AY lineages). Phylogenetic tree was built using Augur utility from the Nexstrain bioinformatic pipeline by employing MAFFT, IQ-TREE and 1000 bootstrap replications (UFBoot).

Variations of clades

According to the GISAID clade nomenclature, the samples in this study could be classified into GH (n = 105, 52%), GK (n = 54, 26.7%), GR (n = 32, 15.8%), G (n = 6, 3%), O (n = 3, 3%) and L (n = 2, 1%). The L clade is an early marker belonging to the MN908947.3 reference sequence and designated for B.56 in this study, while GR and GH clades were previously known as the G clade marked with an S_D614G change that mainly consists of several lineages in the first wave. The GK clade consists of lineages from Delta variants that were dominant in the second wave.

Distribution of lineages

Overall, the lineage distribution did not show a great variation across gender. Exceptions were noted for B.1.1.398 and B.1.470 that was detected in a higher proportion in males (15.5% and 12.9%) compared to females (12.8% and 8.1%), while B.1 and B.1.459 were respectively found higher in females (11.6% and 11.6%) compared to males (10.3% and 6.9%) (Fig. 2C). We found that the majority of genomes during the first wave (date of collection before 16 May 2021, n = 147) was dominated by B.1.466.2 (n = 44, 29.9%). We used 15 May 2021 as the cut-off date between the two waves because the lowest number of new confirmed cases since the beginning of pandemic was recorded on that day, before the number of cases started to increase in the following day. On the other hand, Delta variants were dominant amongst all identified genomes sequenced during the second wave (date of collection 16 May 2021 onwards, n = 53). Those samples mainly consisted of AY.23 (n = 41, 77.4%) and AY.24 (n = 4, 7.5%) which were also described as predominantly Indonesia (https://cov-lineages.org/lineage_list.html). The lineage distribution during the first and the second waves of COVID-19 is shown in Fig. 4A, compared to the official statistics of deaths, recovered and new confirmed cases of COVID-19 in Indonesia (Fig. 4B).

Figure 4 Lineage distribution of SARS-CoV-2 genomes compared to the national statistics.

Side-by-side comparison of lineage distribution is shown across dates of sampling (A) and the official statistics of deaths, recovered and new confirmed cases of COVID-19 (B) at national level (https://covid19.go.id/peta-sebaran), demonstrates two peaks at similar timelines, representing the first and second wave of pandemic.

Key amino acid changes

Notably, a total 199 out of 202 viruses (99.9%) that we sequenced carried the S_D614G and NSP12_P323L non-synonymous mutations. The three genomes that were not found to carry those mutations included two belonging to the lineage B.56 and one from the lineage B.1.416.2, which showed a high N content in the region encoding the spike protein (hCoV-19/Indonesia/JB-BGR-LIPI242/2021, stretches of NNNs 20.66% of overall sequence). Unique changes were found in Indonesian lineages belonging to B.1.1.398 (N_G204R and N_R203K), B.1.470 (NSP_12_P227L) and B.1.466.2 (S_N439K and NSP6_L75F) lineages. A summary of amino acid substitutions is presented in Fig. 5. We also observed that Delta variants carried various amino acid changes that differ from the first wave, which included changes within the spike protein, namely S_T19R, S_G142D, S_E156G, S_F157del, S_R158del, S_L452R, S_T478K, and S_D950N. Additionally, amino acid changes were also found in 22 non-spike protein-encoding genes, namely M, N, NS7a, NS7b, NSP12, NSP13, NSP14, NSP2, NSP3, NSP4 and NSP6S.

Figure 5 Amino acid substitutions among lineages identified in this study.

Only the top 50 substitutions with the highest number of incidences are shown. Several unique substitutions were observed in Delta variants and other variants circulating during the first wave, with the Delta variants having the majority of amino acid substitutions. The highest incidence of substitutions includes S_D614G (n = 199) and NSP12_P323L (n = 197). Two viruses from B.56 lineages were found to carry the least number of mutations (n = 4) and no S_D614G substitution. One virus (hCoV-19/Indonesia/JK-LIPI135/2021) not shown in this figure was found to carry multiple amino acid changes, including S_D614G, S_P681R, S_T478K, S_L452R, NSP6_T77A, NSP3_P1228L, NSP12_P323L, NSP12_G671S, and NSP14_A394V.

Discussion

Whole genome sequencing of SARS-CoV-2 isolates for genomic surveillance has been instrumental in understanding the virus evolution and the dynamics of transmission, as well as monitoring the emergence of mutations (O’Toole et al., 2021; Hadfield et al., 2018; Katoh & Standley, 2016). Such information is critical in aiding public health intervention, the development of vaccines and antivirals, including determining the efficacy of vaccines. This study, which sought to genetically characterize a regional collection of clinical samples from COVID-19 patients using Nanopore sequencer, led to two main findings. First, we observed a shift in the lineages of circulating SARS-CoV-2 in Bogor and surrounding areas, where the Indonesian lineages were dominant during the first wave of pandemic while the second wave was dominated by the Delta variants. Second, we observed unique amino acid substitutions in Delta variants that were different from those found in the first wave, indicating genotypic attributes that may have accounted for higher transmissibility during the second wave of COVID-19, coinciding with the highest record of confirmed cases in provincial and national level at the time.

Like in any other parts of the world, a shift of SARS-CoV-2 variants in a certain geographical area is part of a continuous evolution of the virus to adapt to their environment. Subsequently, new variants may emerge over time as viruses acquire mutations within their genomes, which may improve their fitness and transmissibility (Harvey et al., 2021). Introduction from other areas or imported cases via travelers are also likely to contribute to a high proportion of total COVID-19 cases in many countries (Russell et al., 2021) and subsequently taking over other existing variants, as shown by this study for certain areas in Indonesia. Depending on which genomic changes are more advantageous, competition between different variants may exist and thus certain variants could outnumber others. Additionally, improved host immunity could also affect the proportion of circulating variants. Our study indicated that a variant shift occurred during the two waves in Indonesia, as reflected in the changes in variant dominance. Characteristics of the Delta variants that are more favorable for their survival in the host and transmission could have greatly influenced the dynamics of the different circulating variants between the two waves, resulting in the Delta variants’ dominance in the second wave.

During the first wave of COVID-19 pandemic, up to 15 May 2021, four lineages were predominant in Indonesia, including B.1.466.2, B.1.1.398, B.1.470, and B.1.459. Lineage B.1.466.2 was found in a higher proportion compared to other lineages, accounting for 40% of total viruses published by Indonesian Consortium of SARS-CoV-2 Genomic Surveillance in GISAID (n = 3,732) (Initiative G, 2021) during the first wave of COVID-19. According to the variant graph at Regeneron COVID-19 dashboard (Regeneron, 2021), B.1.466.2 was slowly increasing since November 2020 and reaching the highest peak at the end of March 2021 with 72% of total genomes submitted to GISAID. As such, WHO has labeled this lineage as Variant Under Monitoring on 28 April 2021 (WHO, 2021a) due to some indication that it potentially has higher transmissibility and poses a future public health risk. During this first wave, confirmed new cases at national level peaked up to 14,518 on 30 January 2021 and then slowly declined to the lowest confirmed cases (2,385 cases) recorded on 15 May 2021 (COVID-19 S, 2021).

The presence of the S_N439K and S_ P681R substitutions in the Indonesian lineages are important to note, as these mutations have been shown to confer certain advantages for the variants. The S_N439K mutation has been associated with a similar level of fitness and virulence to the wild-type virus, whilst potentially conferring a stronger binding affinity to human ACE2 receptor and allowing immune escape from antibody-mediated immunity (Thomson et al., 2021). A study employing molecular dynamic simulation further confirmed those attributes and also provided potential mechanisms by which the stronger hACE2 binding affinity and the reduction in efficacy of neutralizing antibodies occur  (Zhou et al., 2021). A distinctive feature of the Delta variant, the S_P681R mutation within the S protein has been correlated to higher fusogenicity, which in turn led to viruses that exhibit enhanced pathogenicity in vivo and greater resistance to neutralizing antibodies compared to the parental virus without this mutation (Saito et al., 2021). Similarly, increased infection due to enhanced cleavage of the S protein has also been associated with the presence of S_P681R mutation, leading to improved viral replication efficiency of Delta variant compared to the Alpha variant (Liu et al., 2021). However, functional assays showed that the S_P681R substitution on its own is not enough to promote enhanced infectivity and transmissibility associated with variants bearing this mutation  (Lubinski et al., 2021). Another substitution at the same position, S_P681H, has also been shown to enhance cleavage of S protein, but did not necessarily result in improved viral entry (Lubinski et al., 2022). Collectively, these studies highlight the importance of these mutations and their potential roles in facilitating enhanced transmission of the Delta variant, which could explain the eventual dominance of the Delta variant in the second wave.

A change in proportion of dominating variants was apparent as the second wave occurred. The Delta variants, including their ancestor B.1.617.2 and AY lineages, began to surpass the Indonesian lineages as they reached 53% by the end of May 2021 and remained as dominant variants towards the end of August–reaching up to 85% of the total genomes from Indonesia submitted to GISAID (June–August 2021, n = 4,189) (GISAID Initiative, 2021). These variants accounted for 91% of total viruses submitted to GISAID with collection date from June to August 2021. According to the officials, a nation-wide record during this second wave amounted to 56,757 new cases on 15 July 2021, followed by a rapid decline to only under 2,000 cases within two months (COVID-19 S, 2021). The dynamics of COVID-19 cases at the provincial level, West Java (COVID-19 PIK, 2021), showed a similar pattern as described at the national level (COVID-19 S, 2021) for both the first and the second waves of pandemic.

Variations in key mutations also followed the shift in dominant variants. For Delta variants, we observed 12 key amino acid changes in spike protein that were mostly similar to other studies, including S_T19R, S_T95I, S_G142D, S_E156-, S_F157-, S_R158G, S_K417N, S_L452R, S_T478K, S_D614G, S_P681R, and S_D950N (CDC, 2021). There is evidence that some of these genetic changes are linked to increased transmissibility (Allen et al., 2021) and reduced effectiveness of vaccines and monoclonal antibodies ((FDA) TUSFaDA, 2021; Deng et al., 2021). On the other hand, substitutions in the region encoding the spike protein in Indonesian lineages mainly consisted of three key mutations, namely S_D614G, S_N439K and S_P681R. Despite the VUM label assigned by WHO, there were no further evidence of either higher transmissibility nor reduced effectiveness of vaccines and therapy linked to the Indonesian lineages.

The most prevalent amino acid change found in our study was the S_D614G, as 199 out of 202 (99%) genomes were found to carry it. This is consistent with the prevalence of this mutation in SARS-CoV-2 genomes in Indonesia (93.1%) and worldwide (97.6%) (Groves, Rowland-Jones & Angyal, 2021). Variants carrying the S_D614G mutation have been suggested to benefit from increased binding affinity to its target ACE2 (Ozono et al., 2021). Although not associated with worse disease severity, the S_D614G change has been implicated in enhanced transmission and higher viral loads (Korber et al., 2020; Plante et al., 2020; Volz et al., 2021). The mutation highly correlates with loss of smell (anosmia) (Von Bartheld, Hagen & Butowt, 2021). Fortunately, a number of experimental and in silico analyses also suggested that the presence of this mutation may not affect the efficacy of vaccines (McAuley et al., 2020). The D614G variant is considered to be G clade by GISAID and B.1 clade by the PANGOLIN (Korber et al., 2020).

Presenting the analysis of genomes obtained using the Oxford Nanopore Technologies (ONT) platform, this study adds to the body of literature that demonstrates the feasibility of Nanopore sequencing for genomic surveillance of SARS-CoV-2, including for the detection of variations within the genome. Studies using a similar platform from ONT showed that nucleotide substitutions (Bull et al., 2020; Li et al., 2020a), as well as deletions within the SARS-CoV-2 genomes (Moore et al., 2020) have also been successfully characterized in samples obtained in the United Kingdom, Brazil and South Africa. Nanopore sequencing also led to the identification of the B.1.1.251 lineage in Brazil in January 2021 (Dos Santos et al., 2021) and eight different lineages of SARS-CoV-2 in South Africa (Engelbrecht et al., 2021). Other advantages that include comparatively lower costs and shorter turnaround time without compromising yield and quality of resulting genomes are attractive features for limited resource settings. Further reduction in turnaround time and costs can also be achieved, as exemplified by a combination use of primer sets that generate 1,200 bp amplicons and the ONT Rapid Barcoding Kit (Freed et al., 2020). Moreover, protocols amenable to high throughput to accelerate sequencing efforts are also available (Baker et al., 2021; Pater et al., 2021). The ongoing improvement in efficiency of cost and time is likely to increase the adoption of Nanopore sequencing of SARS-CoV-2 genomes in many more countries, thereby contributing to a better worldwide genomic surveillance of SARS-CoV-2.

Our study demonstrated a variant shift between two pandemic waves in Indonesia, accompanied by variations in key amino acid changes in genomes of circulating SARS-CoV-2 viruses in Bogor and surrounding areas. The relatively small sampling coverage in a subset of geographic areas may not be a sufficient representation of Indonesia, which would be one of the limitations of this study. However, the regions we selected represent areas with a considerable load of human traffic that potentially influence the transmission dynamics of SARS-CoV-2. More importantly, the results of our current monitoring showcased a similar pattern at provincial and national levels. The current prioritization strategy for sample selection in this study was mainly based on low Ct value to maximise the likelihood of obtaining whole genomes, as at the time we had difficulties obtaining sequencing libraries from samples with Ct values of 30 or above. Further studies would benefit from sample selection based on current local outbreaks, or concerned cases such as reinfection, sporadic cases, as well as those based on severity of illness and cases of vaccinated persons. These considerations are important for a more thorough genomic surveillance in Indonesia, especially in large cities and their surrounding areas, such as Bogor, Jakarta and Tangerang. Nevertheless, we successfully obtained information on Indonesian clade and lineage classifications, as well as the amino acid changes carried by those variants, which would contribute to the much-needed data on viral evolution and transmission in Indonesia.

Conclusions

The second wave of the pandemic in Indonesia witnessed the Delta variants becoming the dominant SARS-CoV-2 variants in Bogor and surrounding areas, taking over the dominance of Indonesian variants during the first wave of pandemic and, reflecting a shift in circulating variants at the national level. Distinct mutations exhibited by the Delta variants compared to other previously identified variants were potentially the reason behind the Delta variants being the cause of the highest average of confirmed COVID-19 cases in the second wave. This was also corroborated by evidence of higher transmissibility of Delta variants that are responsible for massive outbreaks in many countries. In order to prioritise global monitoring and research, and ultimately to inform the ongoing response to the COVID-19 pandemic, whole genome sequencing remains an indispensable tool. Sequencing enables detection of common variants and various amino acid changes in the SARS-CoV-2 genome, contributing to important information on circulating SARS-CoV-2 variants in Bogor and its surrounding areas.

Supplemental Information

File S1 Patients’ metadata

The name of viruses, accession number, qRT-PCR results, date of sampling, demographics as well as lineages, clades and mutations of 202 samples.

Click here for additional data file.

File S2 Whole genome sequences of SARS-CoV-2

All SARS-CoV-2 sequences obtained in this study (n = 202) that have been published to GISAID and are used for phylogenetic tree construction

Click here for additional data file.

We gratefully acknowledge the COVID-19 diagnostic testing team of BSL-3 and BSL-2 Laboratory, National Research and Innovation Agency (BRIN)-Indonesia; Dr. Andika Chandra Putra and Dr. Jamal Zaini from RSUP Persahabatan, Jakarta; and originating laboratories and submitting laboratories as part of Indonesian Consortium of SARS-CoV-2 Genomic Surveillance where genetic sequence data are shared via the GISAID Initiative. We are very grateful for capacity building in the beginning of this project from colleagues at the University of Nottingham: Dr. Susanti Susanti, Dr. Inswasti Cahyani, Prof. Mohammad Ilyas and Prof. Matthew Loose; as part of NICCRAT (Nottingham- Indonesia Collaboration for Clinical Research and Training) partnership.

Additional Information and Declarations

Competing Interests

Author Contributions

Human Ethics

DNA Deposition

Data Availability

The authors declare there are no competing interests.

Anggia Prasetyoputri and Anik B. Dharmayanthi conceived and designed the experiments, performed the experiments, analyzed the data, authored or reviewed drafts of the paper, and approved the final draft.

Syam B. Iryanto conceived and designed the experiments, analyzed the data, authored or reviewed drafts of the paper, and approved the final draft.

Ade Andriani, Isa Nuryana and Andri Wardiana conceived and designed the experiments, performed the experiments, authored or reviewed drafts of the paper, and approved the final draft.

Asep M. Ridwanuloh, Sri Swasthikawati, Hariyatun Hariyatun, Herjuno A. Nugroho, Idris Idris, Indriawati Indriawati, Listiana Oktavia, Yuliawati Yuliawati, Masrukhin Masrukhin, Erwin F. Hasrianda, Linda Sukmarini, Fahrurrozi Fahrurrozi, Nova Dilla Yanthi, Alfi T. Fathurahman, Ari S. Wulandari and Syaiful Rizal performed the experiments, authored or reviewed drafts of the paper, and approved the final draft.

Zahra Noviana analyzed the data, prepared figures and/or tables, authored or reviewed drafts of the paper, and approved the final draft.

Ruby Setiawan performed the experiments, prepared figures and/or tables, authored or reviewed drafts of the paper, and approved the final draft.

Ahmad Fathoni, Wien Kusharyoto, Puspita Lisdiyanti and Ratih A. Ningrum conceived and designed the experiments, authored or reviewed drafts of the paper, and approved the final draft.

Sugiyono Saputra conceived and designed the experiments, performed the experiments, analyzed the data, prepared figures and/or tables, authored or reviewed drafts of the paper, and approved the final draft.

The following information was supplied relating to ethical approvals (i.e., approving body and any reference numbers):

The study protocol was reviewed and approved by the Health Research Ethics Committee, University of Indonesia and Cipto Mangunkusumo Hospital (HREC-FMUI/CMH) (20-10-1321_EXP).

The following information was supplied regarding the deposition of DNA sequences:

All 202 complete genomes of SARS-CoV-2 are available at GISAID: EPI_ISL_1020175-187, 189-191, 193-196, 198-202, 204-206 (n = 28); EPI_ISL_1117450, 452-455, 589, 600, 602-603 (n = 9); EPI_ISL_2840868-894 (n = 27); EPI_ISL_2844812-832 (n = 21); EPI_ISL_2860278-299 (n = 22); EPI_ISL_2860332-349, 360-368 (n = 27); EPI_ISL_3086881-885 (n = 5); EPI_ISL_3186060-061, 064-074 (n = 13); EPI_ISL_3194636 (n = 1); EPI_ISL_3374423-425 (n = 3); EPI_ISL_4004697-704 (n = 8); EPI_ISL_4106122-134 (n = 13); EPI_ISL_518819-820 (n = 2); EPI_ISL_833388-500 (n = 14); EPI_ISL_833503-510 (n = 8); and EPI_ISL_833571 (n = 1).

The following information was supplied regarding data availability:

The qRT-PCR results, demographic patients, lineages, clades, and mutations of 202 samples and the raw data for phylogenetic analysis and all whole-genome sequences are available in the Supplementary Files.

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
