# Peer review of "The dynamics of circulating SARS-CoV-2 lineages in Bogor and surrounding areas reflect variant shifting during the first and second waves of COVID-19 in Indonesia"

_PeerJ, doi:10.7717/peerj.13132_

## Round 0.1 · original submission · Major Revisions

Please address all of the comments by the reviewers.

·

Excellent Review

This review has been rated excellent by staff (in the top 15% of reviews)
EDITOR COMMENT
Thank you for this excellent and most constructive review!

Basic reporting

In this study, Prasetyoputri et al. present genomic surveillance data for SARS-CoV-2 sequences selected from regions in Indonesia. They show that the virus lineage composition changed from the first to second infection wave based on this data set and they describe the virus lineages and mutation profiles detected.

I have some technical questions, suggestions for improvements and also further remarks that should be considered before publication that I mainly list in the “General comments” section.

Sequencing data was uploaded to GISAID. Figures should be provided in higher quality/ resolution for the final publication.

Experimental design

The experimental design is sound. The data set is quite small (202 sequences) but this is also mentioned as a limitation in the paper and it is surely difficult to get samples and perform sequencing in certain areas.

Validity of the findings

The authors present a basic study of SARS-CoV-2 sequencing data that is especially interesting due to the not so well described B.1.466.2 lineage that dominated the first wave. Interestingly, B.1.466.2 also shows the amino acid change P681R in the spike protein which is nowadays also known from overtaking Delta lineages (convergent evolution). Besides, the findings are not surprising in international comparison with Delta dominating in most countries nowadays. However, I really appreciate the effort of the authors because I can imagine that taking samples and establishing such sequencing and analyses systems is not easy in certain areas. Thus, the authors provide insights into the lineage compositions during the two waves in Indonesia, however, their results are also clearly limited by the low sample size of 202 sequences (which is also discussed as a limitation).

On a side note, I checked our German database for SARS-CoV-2 sequences for B.1.466.2 out of curiosity: I can find three sequences of this lineage that were obtained via random sampling in March 2021. All three have the S:P681R and S:N439K changes you also describe. They should also be on GISAID.

Additional comments

[1]
Background: Second sentence, “various”, “variants”, “varying”. You could rewrite this sentence a bit to avoid usage of similar wording

[2]
Objectives: This is not a complete sentence. Maybe “We investigate the ...:”

[3]
I think it is more common to write “S D614G” or “S:D614G” instead of “S_D614G”. However, this might also be a personal preference. Most importantly, it should be consistent throughout the manuscript which it is as far as I checked.

[4]
In this context, it is also not super accurate to write about a “mutation” when actually referring to an amino acid change. For example: “In the spike protein gene, S_D614G and S_P681R mutations ...”. It is more accurate to write “changes” or at least “non-synonymous mutations”. This is also a minor point, because all readers will understand what is meant here by “mutation”. However, you might want to go through the text once more carefully in this context.

[5]
Results: You write that “Most notably, S_T19R, … , and S_D950N mutations were only observed in Delta variants” Why is this so important? These are characteristic changes of Delta and so for me it is not so surprising that they are, for example, not in B.1.466.2.

[6]
Do you have any idea about the importation of Delta to Indonesia? Did they check GISAID sequences in addition to sequences you obtained? Do you have travel metadata associated with your sequenced cases?

[9]
Line 83: “nine large clades” I think you are referring here to the Nextstrain lineages? Or GISAID nomenclature? If so, this should be mentioned.

[10]
Besides the publication about the Pangolin nomenclature system, there is also now a publication about the tool itself which is worth citing.: https://academic.oup.com/ve/article/7/2/veab064/6315289

[11]
Line114: you used Ct already before, consider moving the abbreviation explanation “cycle threshold” some sentences above (Line 109)

[12]
Thanks for the already detailed description of used bioinformatics software and tools. Can you also provide some information on the version of the used ARTIC pipeline? I’m not entirely sure but I think they provide release versions of the pipeline via their GitHub repo. Line 138… “and classified the basecalled reads based on the respective barcodes.” Can you add information if the demultiplexing was done based on single or both barcodes? Demux via both barcodes at the end of an amplicon read is recommended to avoid “barcode bleeding”, however, you will lose some coverage. The option can be set in MinKnow and I’m not sure what the default is in case you used that.

[13]
Methods “Assembly Quality”: What do you mean by “calculated the total raw output base” or “calculate the total raw base”? Do you mean that you calculated the per-base raw-read coverage based on mapping to the Wuhan reference genome? Please be precise. Did you perform any quality-control of the resulting SARS-CoV-2 consensus genome sequences? For example, did you check for the N content? It’s possible that you have some amplicon drop outs which will cause longer N stretches in your consensus sequences. When the N content is high or occurs in important regions like the Spike gene, it gets difficult to reliably call the correct lineage.

[14]
Line 149: “SAM tools” → “SAMtools”

[15]
Line 151: Can you add which version of PangoLEARN you used within PANGOLIN v3.1? Actually, that’s important because the version of PangoLEARN defines which models you used for the machine learning task in lineage assignment. When you re-annotate the lineages of your 202 sequences with the most recent versions of PANGOLIN/PangoLEARN: do the assignments change a lot? I dont’ expect you to redo the whole analyses then, it’s more important to know that your annotations were done with certain models that were up-to-date at the time of analyses but now might have changed.

[16]
Results, Line 158: Can you add here again in which time period these 202 samples were taken?

[17]
Line 160: “A. Patients were male ...” ? is there a number missing here? Something like “160 patients were male ...” Ah, now I see: you refer to Figure 1A and 1B (please remove the dot between “1. A” and “1. B”). However, the Figure 1 in the manuscript does not have subfigures A and B. You mean Figure 2?

[18]
Line 164: “In total, we obtained a complete genome with 20% of total samples were categorized as low coverage by GISAID” Sorry, I can't follow this sentence. Do you mean something like “ Only 20% of our reconstructed genomes were classified as low coverage by GISAID” Although, I’m not convinced that 20% is a low amount here. This also brings me again to my question about the N content of your reconstructed genomes. I think GISAID uses something like a 10% N content cutoff, while others are even more restrictive with e.g. only allowing 5% Ns in the genome.

[19]
Line 167: “Larger coverage depths were obtained with higher total base outputs.” This is obvious, so I suggest removing this sentence.

[20]
Line 173: “B.1470” is this lineage missing a dot? → “B.1.470”

[21]
Line 186: “to the the”

[22]
Line 190: 200/202 had S:D614G, I guess the other two had low coverage in that region or an amplicon drop out? Because (almost) all B lineages should have S:D614G.

[23]
Line 194: “[A] summary of mutation spots”

[24]
You are always talking about “local variants” from the first wave (B.1.466.2, …) and Delta variants from the second wave. Although I think this is the likely scenario, you did not show that what you call “local variants” was not also some introduction from other countries. For sure, it’s likely that Delta was introduced from outside but also that is not shown. So although I get your point to distinguish the lineages of the two waves like that, I suggest rethinking the wording. Sure, the many new mutations in Delta point towards an introduction into the countries. Suggestion: consider changing the corresponding text parts to something like “We also observed that Delta variants carried various mutations that differ from variants detected during the first wave” instead of “local variants”. Or disclaim once that you are referring to variants from the first wave as “local variants” although it’s not shown how they emerged (evolved locally, introduced from outside, ..)

[25]
Line 214: I would argue that in a certain geographical area continuous evolution surely plays one role, but also the introduction from other areas (via travel, …) adds another layer. As stated before, it’s quite likely that the Delta variant was introduced and then took over your “local” variants. And did not evolve in your geographical region how the sentence could be interpreted. I just suggest rephrasing.

[26]
Line 250: I doubt that all these genetic changes are linked to increased transmissibility. Please clarify. E.g. at least weaken the sentence via “There is evidence that some of these genetic changes”

[27]
It is interesting that the local variant you describe B.1.466.2 has S:P681R which then also occurred in Delta (https://outbreak.info/compare-lineages?pango=Delta&pango=Alpha&pango=Beta&pango=Gamma&pango=B.1.466.2&gene=ORF1a&gene=ORF1b&gene=S&threshold=75&nthresh=1&sub=false&dark=true) as you describe. So this points towards convergent evolution at this specific site. I think you could emphasize this point a bit more, which could also explain why B.1.466.2 was successful during the first wave.

[28]
Is there anything known about S:N439K that you describe in B.1.466.2?

[29]
Line 261: “The mutation highly correlates with loss of smell (anosmia).” Is there a study to cite?

[30]
Lines 271-277 I think this can be shortened because information about P.1 or other lineages from Brazil does not really add value to the story.

[31]
Figure 3: It is difficult to distinguish the colors clearly from the small legend. Maybe add for the important lineages again the labels directly into the plot, at least for the higher peaks.

[32]
Figure 4: Phylogenetic tree. The tree and visualization are difficult to interpret and can lead to wrong assumptions. I don’t think that it is a good way to present a selection of lineage representatives in such a tree. For example, AY.* sublineages emerged from the parental B.1.617.2 lineage which is not visible from this tree. Also, certain B.1.* sublineages are descendants of the B.1 lineage, etc.. You only have 202 sequences so I suggest building a complete tree with all of them. It should be feasible to calculate an MSA and tree, e.g. with mafft and IQtree. You could also consider calculating a time tree to add date information from your metadata.

[33]
Please check all Figure labels in the main text. I found some mistakes where the wrong Figure is referenced, e.g. Line 194 “Summary of mutation spots is presented in Figure 4.” which is the tree.

Reviewer 2 ·

Basic reporting

The abstract is concise and summarizes the content of the paper effectively. Within the background portion of the abstract (line 32), specify “highest number of cases of COVID-19” as of what point in time? This will provide some context as to the time period into the pandemic.

Your introduction is very clear and concise, highlighting the advent of the emergence of SARS-CoV-2 and its impact within the population in Indonesia. Some minor grammar edits include proper capitalization in the proper names for SARS-CoV-2 (line 62-63) where it should be “Severe Acute Respiratory Syndrome Coronavirus 2, and COVID-19 (line 63-64) where it should be Coronavirus Disease 2019. For line 68, expand “WHO” into “World Health Organization (WHO)” and clarify when (date, at least month and year) that the total of 245 million cases were detected. This once again gives a concept of time within the pandemic. Minor punctuation edits on “4.7 million” (line 81) where a comma is in place of a period. The general layout of the publication is easy to follow and organized well. Figures, however, seem to be low quality (as in resolution of the images). Zooming in shows high pixilation and should be improved.

It is appreciated that the consensus sequences generated from the sequencing process was submitted into GISAID. However, the raw reads themselves (removed of any reads that align to the human genome i.e., GRCh38, should be deposited into NCBI BioProject for reproducibility of the data collected for the surveillance of COVID-19 in Indonesia.

Experimental design

The methods used are appropriate for long-read sequencing methods. This includes sequencer and bioinformatic pipelines. On line 116, it states that ARTIC v2 and v3 protocols were used “with some modifications” and I would request the direct changes be mentioned immediately followed this statement to be unambiguous and for what reason these modifications were made.

As mentioned before, the sequencing and bioinformatics methods (lines 132-142) are clear with steps and software noted with their associated versions. However, Nanostat (line 148) and “SAMtools” (line 149; correct spelling) are missing their corresponding version numbers. Additional parameters should be included for software analysis for reproducibility. For plots constructed in R (line 154), it would be worthwhile to have the corresponding R code including any preprocessing of data deposited within a Git repository (i.e., GitHub).

Line 164 in your Results section describe a “custom set of multiple primers” whereas in the methods it states “nCoV-2019/V3 primers” (line 119) for SARS-CoV-2 genome amplification. Some clarification is needed here in either the results or methods. If a custom set of primers were used, then a list of sequences should be provided as part of the supplemental information.

Validity of the findings

I appreciate the thorough breakdown of the data, namely examining the lineage distribution across different parameters of the population, even looking variants and sex differences, i.e., which variants infect different sexes more. The general size of the data is sufficient is dependent on both case load and sample quality. It is addressed as a limitation with a note describing population density (line 290-291If you used the ARTICv3 primers on your high Ct samples, was it insufficient to properly recover the genome? Prior evidence suggests it may work even in low virus titer (as seen in Nasir, J.A., et al Viruses 2020;12(8):895.).

Minor edits:
- Line 160 and 162: fix spacing for Figure 1.A and 1.B; perhaps remove the period between them (i.e., Figure 1A and 1B)
- Line 186-187: Wuhan-Hu-1 reference sequence can be replaced with “MN908947.3” as you already referred to this strain (unless you refer to a different accession, in which case please specify)
- Line 191: Small syntax error where it should say “S_D614G” mutation

Additional comments

The paper is a valuable addition, highlighting surveillance of the ongoing COVID-19 pandemic and providing a retrospective look on COVID-19 cases within Indonesia. It is a well written paper with only minor grammatical revisions needed. However, sone clarification on the methods is needed for improved reproducibility including version numbers and namely parameters for the software. Any code used in R should be publicly viewable (this can be alongside your figures) since preprocessing is part of the toolkit. All raw data should be available in the appropriate databases, namely NCBI BioProject for raw reads. Sequences are available in GISAID and provided as supplementary material. The conclusions of the paper highlight the diversity in SARS-CoV-2 lineages and the dominance of the Delta variant over subsequent waves. The value of genome sequencing is reiterated as a continued power in detecting and later responding to the variant (be through monitoring and public health, or research).

---

## Round 0.2 · Minor Revisions

Please address some final small changes.

·

Basic reporting

The language is clear, literature is appropriately cited and raw data (sequence upload GISAID) are shared. Thanks also for providing the R code for plotting.

Experimental design

The experimental design is sound and the bioinformatics methods are appropriately described.

Validity of the findings

The presented findings are clear and supported by the data.

Additional comments

Thanks for the thoughtful revision. I am satisfied and my raised points were addressed. I just have two minor suggestions:

1) Please check the sentence
l314-315
"A study employing molecular dynamic simulation [study] further confirmed ... "
Should the last [study] be deleted?

2) Tree, Figure 3
Thanks, this tree is much better than the previous one! But I recommend removing the labels at the nodes. They are not readable anyway. Also, you could maybe color also the branches and not only the nodes to it's better visible which parts of the tree correspond to which lineage. I can recommend https://www.iroki.net/ if you have the newick tree file available. You can also provide a metadata file to easily color the lineages differently.

Thanks!

Reviewer 2 ·

Basic reporting

No comment.

Experimental design

No comment.

Validity of the findings

No comment.

Additional comments

All my previous concerns have been dealt with accordingly or explained!

---

## Round 0.3 · accepted · Accept

Final suggestions have been addresssed.